# Degradation of Cellulose and Hemicellulose by Ruminal Microorganisms

**DOI:** 10.3390/microorganisms10122345

**Published:** 2022-11-27

**Authors:** Paul J. Weimer

**Affiliations:** Department of Bacteriology, University of Wisconsin, Madison, WI 53706, USA; pjweimer@wisc.edu; Tel.: +1-608-770-6071

**Keywords:** carboxylate platform, cellulose, fiber, hemicellulose, rumen, volatile fatty acids

## Abstract

As major structural components of plant cell walls, cellulose and hemicellulose are degraded and fermented by anaerobic microbes in the rumen to produce volatile fatty acids, the main nutrient source for the host. Cellulose degradation is carried out primarily by specialist bacteria, with additional contributions from protists and fungi, via a variety of mechanisms. Hemicelluloses are hydrolyzed by cellulolytic bacteria and by generalist, non-cellulolytic microbes, largely via extracellular enzymes. Cellulose hydrolysis follows first-order kinetics and its rate is limited by available substrate surface area. Nevertheless, its rate is at least an order of magnitude more rapid than in anaerobic digesters, due to near-obligatory adherence of microbial cells to the cellulose surface, and a lack of downstream inhibitory effects; in the host animal, fiber degradation rate is also enhanced by the unique process of rumination. Cellulolytic and hemicellulolytic microbes exhibit intense competition and amensalism, but they also display mutualistic interactions with microbes at other trophic levels. Collectively, the fiber-degrading community of the rumen displays functional redundancy, partial niche overlap, and convergence of catabolic pathways that all contribute to stability of the ruminal fermentation. The superior hydrolytic and fermentative capabilities of ruminal fiber degraders make them promising candidates for several fermentation technologies.

## 1. Introduction

Ruminant animals constitute a mammalian order that includes some of humankind’s most important livestock species (cattle, sheep, goats and buffalo), as well as numerous wild animal species. Ruminants are exquisitely evolved organisms that have harnessed a community of microbial symbionts to facilitate an unusually effective bioconversion of plant biomass to meet their extensive nutritional needs [1,2]). From a nutritional standpoint, the variable degradability of cellulose and its oft-underappreciated cousin, hemicellulose, places them between pectins and cell solubles (rapidly and completely degraded) and lignin (indigestible). As a result, cellulose and hemicellulose degradation represent the greatest source of variation in the digestibility and nutritional value of ruminant diets, and thus is a central target for improving animal agriculture. In addition, examination of the ruminant’s impressive ability to utilize relatively recalcitrant biomass can offer important insights to improve industrial biorefinery processes aimed at bioconversion of cellulosic feedstocks.

Although the biodegradation of cellulosic biomass is ultimately an enzymatic process, this biodegradation cannot be fully understood without considering the process at the cellular (as opposed to the subcellular/enzymatic) level. Such a cell-level analysis, first advocated two decades ago [3], is particularly relevant for the ruminant system, where cellulolytic microbes are particularly dependent on adherence to the degradable surface, and the concentration and activity of free (unbound) cellulolytic enzymes is minimal.

This review will consider the role of microorganisms (as opposed to free enzymes) in the degradation of both cellulose and hemicelluloses, which are physically associated with one another within the plant cell wall matrix. As will be seen, the degradation of these two polysaccharides displays some similarities, but numerous differences; these differences have important implications in both ruminant nutrition, and in the potential use of ruminal microbes in industrial cellulosic bioprocessing. For the purpose of this review, these two components will be collectively referred to as “fiber”, although this latter term is usually more widely used for the entire plant cell wall matrix, which also includes lignin, pectin and some other constituents. Plant cell wall architecture and tissue organization are complex, and are highly variable across plant taxa [4]. This greatly influences the ability of fibrolytic microbes to access their substrates. However, this review will mention these effects only in passing, and will instead focus mostly on the degradation of the cellulose and hemicellulose polymers themselves.

## 2. The Ruminant Digestive System

Across the ruminant order, there is great variation in feeding behavior that results from a complex interplay of animal anatomy, available feed type, and evolutionary history, as superbly described by Van Soest [5]. Regardless of this variation, it is clear that digestion in the ruminant features two unique evolutionary innovations: (1) the presence of an enlarged, thermoregulated (~39 °C) forestomach (the rumen) that harbors an interactive anaerobic microbial community that ferments feeds to volatile (short-chain) fatty acids (VFA); and (2) rumination, a periodic regurgitation and mastication of fermenting feed material (“chewing of the cud”) that maximizes the rate and extent of feed conversion.

The rumen typically accounts for about 15% of the animal’s weight, and its large volume serves to increase the retention time of feeds, thereby allowing more extensive feed conversion. Retention times vary with animal species, individual animal, feed type and measurement method, but are typically 2–3x longer for the solid phase than for the liquid phase (e.g., in dairy cows, 8–20 h for liquid, and 28–65 h for digesting solids; [6,7,8]). In addition, the rumen contains a floating mat of larger feed particles whose retention times are even longer, which permits the persistence of some slow-growing community members (the protists) that would otherwise be washed out to the lower intestinal tract [9].

The rumination process involves the regurgitation of a bolus into the buccal cavity, where the animal captures the larger particles for rechewing while squeezing away most of the liquid and smaller particles for immediate return to the rumen. This step is followed by mastication of the captured feed, employing a uniquely shaped dentition and a rotary motion of the jaw that results in a shearing and delamination (rather than a cutting) of the plant material [10]. This has the overall effect of greatly increasing the exposed surface area of the substrate (the rate-limiting factor in biodegradation). Because the energy-intensive costs of cutting are avoided, and because the feed has been pre-wetted in the rumen, the energy costs for mastication are low, typically only a few percent of the gross energy of the feed [11]. This low energy requirement enables rumination to occupy a large fraction of a ruminant’s daily routine (up to 10.5 h d^−1^ in lactating dairy cows, with longer times associated with higher fractions of plant cell wall material in the diet [12]).

Once rendered accessible to the microbial community, feed components are subjected to anaerobic fermentation by a complex assemblage of microbes, yielding as final products VFA containing 2 to 6 carbon atoms, in relative amounts that decrease with increasing chain length [1]. These VFA serve as the main sources of energy and anabolic precursors for the host animal, which is unable to directly use dietary carbohydrates. In addition to the VFA, gaseous products (carbon dioxide and methane) are side products of the fermentation, and are released by the animal via eructation.

## 3. Ruminal Cellulolytic Microbes

Cellulolytic microbes are of central importance in nutrition of the host, and their very early appearance in newborn ruminants correlates with a rapid development of the rumen itself, and its capacity to exploit highly fibrous diets [13,14]. Our knowledge of the ruminal cellulolytic community has been derived from two sources: classical enrichment and isolation of species into pure culture; and more recent “culture-independent” characterization of the bulk ruminal community using small-subunit rRNA gene sequencing and metagenomic techniques. Ruminal cellulolysis has been ascribed to three major microbial groups—bacteria, protists and fungi—though the relative contribution of each group has been a source of controversy [15,16]. 

### 3.1. Bacteria

Three bacterial species—*Fibrobacter succinogenes*, *Ruminococcus albus* and *Ruminococcus flavefaciens*—have long been considered to be the predominant agents of cellulolysis in the rumen [1,2]. Quantitative studies using 16S rRNA abundance in bacterial community DNA indicate that these species appear to be universally present in all ruminants examined, sometimes at several percent of the bacterial community. In dairy cows, *F. succinogenes* and *R. flavefaciens* have similar population sizes, while that of *R. albus* is typically around an order of magnitude lower (Table 1) [17,18,19,20,21]. The three species have several common characteristics—they are strictly anaerobic, nonmotile, and unable to produce endospores or other resting cell structures. All three can obtain virtually all of their N from ammonia; require low concentrations of C_4_–C_5_ branched chain fatty acids for growth, and are very acid-sensitive (unable to initiate growth at pH < 6). 

For all three species, adherence to cellulose is a prerequisite for rapid cellulolysis (Figure 1). Adherence is thought to provide several benefits, including direct contact of surface-bound cellulases with the substrate (thus minimizing wasteful release of cellulases into the bulk liquid phase); preferential access to cellodextrin products of cellulose hydrolysis (which can also be utilized by many non-cellulolytic species); and longer retention time of cells in the rumen, enhancing contact time between cells and substrate [3]. The importance of adherence is revealed by observations that more rapid growth of ruminal microbes results in a greater adherence of cells to cellulose. For *F. succinogenes* growing in cellulose-limited chemostats, the percentage of cells adhering to cellulose increased linearly with increasing dilution rate (i.e., increasing growth rate), from 18% at D = 0.02 h^−1^ to 69% at D = 0.07 h^−1^ [22]. Moreover, non-adherent mutants of *F. succinogenes* have been reported to hydrolyze cellulose poorly, and are unstable (i.e., they often revert to an adherent phenotype, and regain the ability to hydrolyze crystalline cellulose [23]). Adherence is also important for the cellulolytic ruminococci. *R. flavefaciens* is unable to degrade cellulose when incubated with methylcellulose, a soluble cellulose ether that prevents adherence of cells to cellulose, and does not inhibit growth on soluble sugars [24]. *R. albus* requires a specific growth factor, 3-phenylpropanoic acid, for growth on cellulose; this factor, which is not required for growth on soluble sugars, reportedly enhances the affinity of cells for cellulose [25,26].

Despite the above shared attributes, the three predominant cellulolytic bacterial species differ in numerous respects, including phylogenetic origin, types and organization of cellulolytic enzymes, mode of adherence to substrate, fermentation end products, and quantitative growth properties [27,28,29] (Table 2). At least two species (*F. succinogenes* and *R. flavefaciens*) also differ in their relative rates of hydrolysis of different cellulose allomorphs (i.e., cellulose fibers having different crystallite dimensions and hydrogen bonding patterns; [30]). 

The huge variety of carbohydrate active enzymes (CAzymes) among ruminal fibrolytic bacteria underlie many of the differences among the cellulolytic species, and even differences among strains of a given species. While a few generalities in enzyme diversification are mentioned below, enzymological details are beyond the cell-based focus of this review. For a more detailed discussion of these details, the reader is referred to the excellent review by Moraïs and Mizrahi [14].

#### 3.1.1. Fibrobacter Succinogenes

*F. succinogenes* (originally *Bacteroides succinogenes*) is a Gram-negative specialist cellulose-degrading bacterium, and is one of only two described members of the phylum Fibrobacteriota [31,32]. Although the Fibrobacteriota have long been regarded as GI tract microbes, more recent work has identified them as residents of other habitats such as sediments and landfills [33]. While *Fibrobacter* is capable of hydrolyzing cellulose and a variety of hemicelluloses, only the hydrolytic products of cellulose support growth, leading to speculation that hemicellulose hydrolysis by this species merely serves as a means of increasing access to cellulose within the plant cell wall matrix [34]. Adherence to cellulose is mediated by slime proteins (“fibro-slime”) and pilins that permit unusually close contact to the cellulose [35]. 

*F. succinogenes* produces a variety of cellulases that are secreted extracellularly using type II and III secretion systems [34]. However, enzyme titers are remarkably low for an actively cellulolytic organism. Heterologous expression of *F. succinogenes* cellulases in *E. coli* also yields cellulolytic activity far inferior to that of native *F. succinogenes* [36]. These observations suggest that most of the enzymatic activity occurs at the cell-cellulose interface of adherent cells. Recent detailed studies by Raut et al. [36] provide strong evidence that *F. succinogenes* secretes cellulolytic enzymes through the peptidoglycan cell wall, upon which they are covalently bonded to lipid A of the outer membrane to form a complex with other surface components, including fibro-slime proteins. There, in concert with carbohydrate-binding modules (CBMs), cellulose hydrolysis yields cello-oligmoers (cellodextrins). While cultivation on cellulose enhances the expression of a number of cellulase and hemicellulase genes (relative to cultivation on glucose), the mechanism of this enhancement is not clear-cut, as chemostat studies with both *F. succinogenes* [37] and the nonruminal *Clostridium thermocellum* [38] have shown that expression of a large number of genes, including many involved in cellulose metabolism, is controlled by growth rate rather than by substrate type (cellulose vs. soluble sugars). Consequently, many transcripts important in cellulose degradation can be expressed by growing the bacteria on cellobiose at low dilution rates. This confounds direct comparison of batch cultures on cellulose vs. on soluble sugars in many studies conducted in batch mode, because the two cultures were grown at very different rates.

An additional unique feature of cellulose degradation by *F. succinogenes* is the arrangement of adherent cells along the crystallographic axis of cellulose (Figure 1), leading to the formation of parallel groves in the cellulose fiber that can be revealed by removal of adherent cells using methylcellulose [39] or an undefined agent in extracts of the legume *Astragalus cicer* [40]. This ordered arrangement of cells is not observed when cells adhere to amorphous (non-crystalline) cellulose [41]. The means by which the bacterial cells align themselves along the crystallographic axis of cellulose remain to be elucidated, but likely involves an ordered positioning of cellulolytic enzyme complexes on the cell surface in the absence of a cellulosome structure. This hypothesis is strengthened by inspection of the unique suite of CAzyme genes in the *F. succinogenes* genome; these encode for 31 known cellulases distributed into CAzyme families GH5, GH8, GH9, GH45, and GH51), along with 10 unique fibro-slime proteins and seven noncatalytic proteins that contain CBMs; two of these contain CBMs known to bind to cellulose [34]. Interestingly, the genome does not encode any readily identifiable processive β-1,4-exoglucanase, an enzyme type widely considered essential for rapid degradation of crystalline cellulose, further attesting to the novelty of the *F. succinogenes* cellulolytic system.

*F. succinogenes* produces succinate as its major fermentation product, aided by PEP and pyruvate carboxylases that fix CO_2_ into three-carbon fermentation intermediates of glucose catabolism to yield C_4_ dicarboxylic acids, including succinate, its major fermentation product [34]. Other fermentation products include acetate and formate, but not higher VFA, ethanol or H_2_. Chemostat studies indicate that fermentation product ratios appear to be relatively similar across growth rates and culture pHs [27].

#### 3.1.2. Ruminococcus

Ruminococci are members of the Gram-positive phylum Firmicutes. Two members of the genus (*R. albus* and *R. flavefaciens*) are active hydrolyzers of both cellulose and various hemicelluloses. Unlike *F. succinogenes*, these ruminococci can ferment several types of hemicellulose, including xylans, glucomannan and lichenan, but not arabinogalactans or storage glucans such as β-1,3-glucan, laminarin, or starches [42]. Like several non-ruminal cellulolytic members of the Firmicutes (e.g., *Clostridium thermocellum*), many strains of cellulolytic ruminococci are highly pigmented (yellow or orange), and the intensity of pigmentation roughly correlates to cellulolytic ability.

Adherence to cellulose by *Ruminococcus albus* is mediated via both pilin-like proteins and an extracellular glycocalyx composed of glucosyl, mannosyl and xylosyl residues (a mixture unusual among EPS-producing bacteria), along with a substantial protein content (Christopherson et al., 2014) [42]. *R. flavefaciens* also produces an extracellular glycoprotein coat, but its carbohydrate component is largely composed of rhamnosyl, glucosyl and galactosyl units [43]. 

Comparative genomics of *R. albus* 7, *R. albus* 8, and *R. flavefaciens* FD-1 have revealed that these three strains share 1,234 orthologs, 55 of which were predicted to encode for CAZymes. However, nearly half of the predicted CAZymes were shared with noncellulolyic ruminococci. *R. albus* 7 contains genes for the assembly of cellulosome complexes, and also contains about a dozen genes for cellulase, endo-xylanase and fibronectin-containing proteins that contain a novel CBM37 binding domain (unique to this species) [44]. Transcription of these genes is enhanced at least 4-fold when cells are grown on cellulose versus on cellobiose in chemostats at identical growth rates [42] to avoid the above-mentioned confounding effects of growth rate on gene expression. In contrast to strain 7, *R. albus* strain 8 does not contain genes for scaffoldin proteins and most of its GHs lack dockerin domains, indicating that the cellulolytic system in this strain is non-celluosomal [45].

*R. flavefaciens* FD-1 produces a very elaborate cellulosomal complex [46] that can potentially involve up to 223 separate dockerin-bearing ORF products available for insertion onto the cellulosomal scaffoldin. The bacterium further displays its uniqueness in that integration of enzymes into the cellulosome assembly involves single-binding dockerins attached to the scaffoldin-bound cohesion modules (novel for cellulosomes [47]), while a more conventional dual-binding mode is involved in cellulosome cell-surface attachment. Additionally, this strain also produces a large number of GHs that lack dockerin domains, suggesting that they may function as free extracellular enzymes [14].

#### 3.1.3. Other Cellulolytic Bacteria

The early rumen microbiology literature contains several reports on novel cellulolytic isolates that have since faded into obscurity, if not oblivion. Perhaps the most interesting of these is *Clostridium lochheadii* [48], a sporeformer that—unique among highly cellulolytic bacteria—was also highly proteolytic, and which gradually lost both sporulation ability and culturability in the years following its isolation. Later, Hungate [1] described this species as “the most rapid cellulose digester isolated from the rumen”, superior even to *F. succinogenes* and the ruminococci. Unfortunately, this species is no longer in culture. 

Also lost from culture was another contemporary isolate, *Clostridium longisporum*, which was substantially less cellulolytic but highly xylanolytic and was notable for its intense orange colony pigmentation [49]. This latter trait was exploited to quantify the bacterium’s population density in the one of the earliest experimental attempts at establishing a bacterial population in the rumen by direct dosing [50]. 

A third species, *Eubacterium cellulosolvens*, (originally *Cillobacterium cellulosolvens*) is of interest due its production of lactate and butyrate as its major fermentation products -- atypical for cellulolytic bacteria [1]. A neotype of this species was later isolated by Van Gylswyk and Van der Toorn [51]. Its product mix of lactate, acetate and butyrate would likely be useful as chain elongation substrates for the carboxylate platform (see Section 3.9.1), Recent genomic studies suggest that this species may merit classification into its own novel genus within the family Lachnospiraceae [52].

Even within cellulolytic species, certain previously described strains of particular interest appear to be no longer available. A prime example is *Ruminococcus albus* Ce63, which reportedly was grown in continuous culture on cellulose at dilution rates of 0.17 h^−1^ [26], nearly double the maximum attainable by other strains of this species.

#### 3.1.4. Bacterial Utilization of Hydrolytic Products

The major products of ruminal cellulose hydrolysis are not glucose (G_1_) or cellobiose (G_2_), but cellodextrins (cellotriose through cellohexaose [G_3_-G_6_]) [3], and growth experiments reveal that all three predominant cellulolytic species have higher affinities (lower K_s_ values) for these cellodextrins than for cellobiose or glucose [53]. This observation is consistent with the notion that these compounds are more beneficial substrates because they provide greater net energy yield by virtue of energy savings accrued from intracellular phosphorolytic cleavage following uptake ([Glc]_n_ +P_i_ → [Glc]_n−1_ + Glc-1-P) [3]. In *F. succinogenes*, it appears that growth yields on glucose are further reduced by an unproductive, simultaneous synthesis and degradation (“futile cycling”) of glycogen [54]. The cellodextrin phosphorylase reaction is reversible, and although equilibrium lies in the direction of catabolism (decreasing cellodextrin chain length), under certain circumstances the reaction can run in reverse, to synthesize longer cellodextrins [22]. 

Cellulose-limited chemostat cultures of *F. succinogenes* display complete colonization of cellulose particles by adherent cells, as well as a substantial density of planktonic cells that presumably are unable to gain access to the fully colonized substrate [22]. A proportion of these planktonic cells show active cell division, suggesting that they are growing on soluble products of cellulose hydrolysis, confirming earlier studies that demonstrated rapid glucose metabolism in non-adherent resting cells [55]. These observations suggest that products of cellulose hydrolysis are incompletely captured by adherent cells, despite their close adherence to cellulose. Cellulose-limited chemostat cultures of ruminococci display similar behavior. In view of this apparent escape of cellodextrin products, it is surprising that all three cellulolytic species adhere to cellulose as a monolayer [39,56], as opposed to multiple layers found in typical bacterial biofilms that develop on both degradable and inert surfaces. 

Once hydrolytic products are metabolized to G-1-P, they are typically fermented via the Emden–Meyerhoff pathway to the central intermediate pyruvate, from which different species produce their species-specific mixture of fermentation end products (Table 2).

### 3.2. Protists

The rumen contains a diverse range of anaerobic protists (often referred to as protozoa). Members of the group have well-established functions in the predation of bacteria, or the engulfment, sequestration and fermentation of starch. They also have a role in fiber degradation, though study of this capability is complicated by the difficulty of maintaining ruminal protists in culture [57,58], and by the possible presence of endosymbiotic bacteria, some of which can potentially degrade cellulose [14]. 

Much of our understanding of the in vivo role of ruminal protists is derived from defaunation studies, in which the protist population is removed temporarily by feeding of saturated C_10_–C_14_ fatty acids or various other chemical agents [59]. Defaunated ruminants typically show greater feed efficiency and lower methane emissions than their faunated counterparts. They also show significant reduction in fiber degradation. In a 23-study meta-analysis conducted by Newbold et al. [60], digestibility of neutral-detergent fiber (NDF, which contains cellulose, hemicellulose and lignin) and acid detergent fiber (ADF, which contains cellulose and lignin but no hemicellulose) in defaunated animals was reduced on average by 20% and 16%, respectively, although there was some compensatory recovery of fiber digestibility in the hindgut.

An active role in fiber degradation is consistent with the protists’ preferential localization in the floating mat within the rumen, which is the site at which fiber concentration is maximal by a large margin. The ability of the protists to “hide” within the ruminal mat reduces their rate of passage to the lower tract [61], and allows them to persist and to degrade fiber despite their rather leisurely growth rates (e.g., generation times of 11.3 h for *Eudiplodinium impalae* and 18.7 h for *Enoploplastron triloricatum* [62]). Degradation appears to involve some enzymatic attack on plant particles prior to eventual engulfment and intracellular digestion [63].

The greatest capacity for cellulose degradation among the protists is found in the ciliate group. For example, addition of cellulose to suspensions of the ciliate protist *Diploplastron* increased cell numbers in vitro, and cell extracts of this species hydrolyzed a wide range of polysaccharides, including cellulose, CMC, murein, starch and chitin [64], although rates of cellulose hydrolysis were modest. Similarly, the ciliate protist *Epidinium ecaudatum* was reported to engulf particles of microcrystalline cellulose (MCC) and—much less frequently—xylan. In vitro incubation of this protist with MCC increased cell density, and cell-free extracts hydrolyzed both CMC and xylan [65]. More recently, this species has been shown to produce a cellobiohydrolase [66]. Some ciliate protists engulf cellulose particles more rapidly than they do starch particles and, interestingly, convert cellulose hydrolysis products to an amylopectin at in vitro rates of 0.4 to 4.75 μg min^−1^ (mg protein)^−1^ [67]. 

### 3.3. Fungi

Orpin [68] revolutionized our concept of the rumen microbiome by discovering that certain motile eucaryotes, frequently observed under light microscopy and thought to be protists, are actually the zoospore stage of a new group of fungi now classified in the phylum Neocallimastigomycota. These fungi, which represent one of the earliest evolutionary divergences from a least common eucaryotic ancestor, have a complex life cycle. Motile zoospores (which lack a cell wall and thus are also capable of amoeboid movement on solid surfaces) attach to degradable plant tissue, typically within 0.5 h of release from the fungal sporangium; they then lose their flagella, and metamorphose into a nonmotile thallus that displays filamentous vegetative growth, eventually producing a sporangium filled with the next generation of motile zoospores [69]. The essentiality of adherence to fiber is shown by the fact that methylcellulose simultaneously inhibits both adherence to, and degradation of, cellulose [70]. Moreover, treatment with methylcellulose of cultures growing on cellulose detaches cells from the fibers and arrests cellulose degradation.

The Neocallimastigomycota mycelia produce a host of fibrolytic enzymes, including cellulases that are organized into cellulosome-like organelles on the cell surface. Metabolically, the ruminal fungi resemble some of the ruminal ciliate protists in that they are strictly anaerobic and have a fermentative energy metabolism, and are difficult to maintain in culture [71]. Fermentation balances reveal a host of end products, including acetate, formate, ethanol, lactate (both D- and L-isomers), succinate, H_2_ and CO_2_ [72]. Carbon recoveries in fermentation products are high (94–102%), suggestive of low cell biomass yields that may contribute to their poor growth in vitro.

Though some of the ruminal fungi produce cellulases of very high specific activity, much of the fungal plant-degrading capacity appears to stem from the physical force exerted by growing hyphal tips (appressoria) within degrading plant tissues. The resulting fracture and splitting of plant tissues allows invasion by other degradative microbes, particularly bacteria. These features give the ruminal fungi an outsized influence in plant biomass degradation. 

Estimates of fungal abundance in the rumen vary widely. Early studies by Orpin [68] suggest that they comprise up to 8% of rumen microbial biomass. On the other hand, Rezaeian et al. [73] suggested a value of 20% of microbial biomass, based on chitin content in the rumen and estimates of chitin abundance in the fungal cell, combined with estimates of bacterial and protist biomass. This latter value is at odds with the relative scarcity of fungal mycelia under direct microscopic observation of ruminal contents.

### 3.4. Kinetics of Ruminal Cellulose Degradation

Having now identified the microbial players in the cellulose-degrading ruminal community, we can turn to the quantitative aspects of cellulose degradation. A number of studies have examined the kinetics of ruminal cellulose degradation. Those in which pure cellulose was used as substrate have almost exclusively been conducted in vitro, either in batch or continuous culture. Because in vitro cultivation leads to rapid loss (within hours) of the eucaryotic microbial community (both protists and fungi), in vitro data reflect almost entirely the activity of its bacterial component. The fact that kinetic behavior in the rumen itself largely reflects in vitro data is testament to the central role of bacteria in ruminal cellulose digestion, either directly or in their partial capacity for niche replacement of the displaced eucaryotes.

Several studies, both in vitro and in rumine, have demonstrated that ruminal cellulose degradation is a first-order process with respect to available cellulose concentration. After an initial lag period, the rate of disappearance of pure cellulose is directly proportional to cellulose concentration [74], and is also directly proportional to the accessible surface area of cellulose particles [75]. In the case of more natural cellulosic biomass such as grasses and alfalfa, first-order kinetics for the polysaccharide fraction is also revealed after correcting weight-loss curves for the amount of indigestible lignin in the residue [76]. By contrast, the concentration of ruminal microbes has little effect on in vitro cellulose degradation, as revealed by the fact that ruminal inoculum can be diluted approximately 6-fold before the first-order rate constant begins to decline [77]; as would be expected, lower inoculum levels further decrease the first-order rate constant and increases the lag time prior to the onset of degradation [78]. Maximal rate constants for the degradation of pure cellulose by mixed ruminal inocula, determined by weight loss measurements, are generally around 0.1 h^−1^ [75,76,79] (Table 3). Rate constants for cellulose degradation by pure cultures of the predominant cellulolytic bacteria, determined from fractional cellulose disappearance at steady state in continuous cultures at different dilution rates, are also generally around 0.1 h^−1^ (Table 2).

Though the first-order kinetics of cellulose degradation in the rumen is well-established, its implications are often ignored: Cellulose degradation is limited by the substrate availability, not by the amount or activity of cellulase enzymes, or the density of cellulolytic microbes. Thus, ruminal cellulose degradation cannot be enhanced by further addition of cellulase enzymes or cellulolytic microbes (including those engineered for cellulase overproduction) – even if such microbes could establish themselves in the highly competitive ruminal habitat.

Ruminal cellulose degradation displays considerable dependence on pH. Russell and Dombrowski [80] noted that pure cultures of the predominant ruminal cellulolytic bacteria cannot initiate growth at pH < 6. Slyter [81] used continuous culture of a mixed ruminal inoculum to select for improved cellulose degradation at pH 5.0 or 5.5, but the resulting communities, at a pH of 5.5, showed no improvement in the weak cellulose degradation observed by cultures selected at pH 6.5. Mouriño et al. [77] reported that in batch cultures of mixed ruminal bacteria, the first-order rate constant for Sigmacell 50 MCC disappearance was maximal at an initial pH 6.8. In fact, once cellulose degradation began, it continued at a fixed rate constant that depended on initial pH, but this rate constant declined linearly with initial pH before reaching zero at pH 5.3. Rate constants paralleled the level of cellulose-adherent P and N (proxies for cell adherence, as the pure cellulose substrate lacked both P and N). The data suggest that adherence precedes and is required for cellulose degradation, and that initial pH is the primary driver for pH effects. Once established, cellulose degradation can continue even at low pH, until the a minimum pH is reached at which cells detach from the cellulose, Hu et al. [82] subsequently examined pH effects on the mixed bacterial community in a series of bioreactors having an initial concentration of Avicel PH102 (another MCC) of 10 g/L, with pH control at 0.5 pH unit intervals between 4.8 and 7.3. Optimal cellulose degradation at all time points (8–120 h) was observed at pH 6.8, and cellulose conversion declined rapidly at pH values under 6, with no degradation at pH 4.8. Overall, the reported pH effects observed in vitro are consistent with the pH profile of the rumen itself: pH in grazing or roughage-fed animals rarely decreases below 6, and feeding diets high in grains or concentrates results in acute or subacute acidosis accompanied by substantially reduced extents of fiber degradation [2].

An interesting feature of cellulose degradation kinetics by mixed ruminal microbes is that there is a substantial log period (7–17 h) prior to cellulose degradation becomes measurable [75,77,82]. This lag period is observed both in vitro and in sacco (i.e., in mesh bags containing cellulose and incubated within the rumen for various time periods). By contrast, lag times for cellulose within intact forages are one-third to one-half those on pure cellulose [83]. This suggests that microbial contact with and adherence to fiber – the initial steps in initiating fiber degradation—is dependent on the chemistry or structure that fiber, and that the microbes preferentially recognize and adhere to a non-cellulose component of the fiber, perhaps one or another hemicellulose.
microorganisms-10-02345-t003_Table 3Table 3First-order rate constants for degradation of cellulose by microbial communities from the rumen and from mesophilic anaerobic digesters.InoculumSubstrateInitial Concentration (g L^−1^)k (d^−1^) ^c^ReferencesMixed ruminalAlfalfa cellulose
1.46–2.45[74]Mixed ruminalWhatman filter paper
1.68[79]Mixed ruminalCotton
0.96[79]Mixed ruminalSigmacell 50 MCC ^a^101.85[75]Mixed ruminalSigmacell 50 MCC102.40[77]Sludge digesterND ^b^
0.04–0.13[84]Sludge digester“Cellulose powder”50.94[85]Sludge digesterFilter paper20.247[86]Sludge digesterND ^b^
0.1[87]Sludge digesterND ^b^
0.066[88]Sludge digesterSigmacell 20 MCC200.252[89]^a^ MCC, microcrystalline cellulose. ^b^ ND, not described. ^c^ First-order rate constant. Because of their more rapid rates, ruminal data are typically expressed on an hourly basis, for which the data here range from 0.04 to 0.108 h^−1.^


### 3.5. Quantitative Comparison of Ruminal versus Non-Ruminal Cellulose Degradation

An essential feature of biomass utilization by the ruminant is its high productivity, underpinned by a rapid rate of cellulose degradation. The superior degradative capability of the rumen microbial community on cellulosic biomass (compared another familiar degradative process, anaerobic digestion (AD)) can be inferred from comparison of its first-order rate constants for cellulose degradation, which are typically 10–20× those in a mesophilic (35–40 °C) anaerobic digester [74,75,77,79,84,85,86,87,88,89] (Table 3). A notable exception was found in the AD study of Noike [85], but in this latter study cellulose removal rates were calculated by difference, and at the shortest solids retention time only ~2% of the original 5 g cellulose L^−1^ were removed, a fraction subject to large measurement errors. The range of rate constants in the other AD studies are roughly similar to the range over 5 studies that used a variety of industrial paper feedstocks, reviewed by Gonzalez-Estrella et al. [90].

To express this difference between ruminal and AD cellulose fermentations in another way, consider a cow consuming 20 kg of fresh pasture per day on a dry matter (DM) basis (e.g., see [91]). Assuming a cellulose content of 20% of DM, and a ruminal digestibility of 50%, cellulose degradation 

=20 kg forage DM × 0.2 kg cellulose (kg forage DM)^−1^ × 0.5 g cellulose degraded (g cellulose)^−1^

=2.0 kg cellulose degraded d^−1^.

At a working ruminal volume of 70 L, the volumetric rate of cellulose degradation 

=(2000 g cellulose d^−1^)/(70 L × 24 h d^−1^) = 1.19 g L^−1^ h^−1^.

In addition, using an estimated cell concentration of 10 g of microbial cells L^−1^, [92], specific rate of cellulose degradation = (1.19 g L^−1^ h^−1^)/(10 g cells L^−1^) = 0.119 g cellulose degraded (g cells)^−1^ h^−1^.

Volumetric rates of cellulose degradation during mesophilic AD are surprisingly sparse in the literature. Bolanji and Dionisi [89] reported a maximum cellulose degradation rate of 0.17 g L^−1^ d^−1^ in a lab-scale anaerobic digester fed Sigmacell 20 MCC at 20 g L^−1^ concentration, at residence times of 20–80 d. Donaldson and Lee [93] achieved much faster rates, up to 1.34 g L^−1^ d^−1^ in a 55-L digester fed mixed paper (shredded to 1 mm) at 5–50 g L^−1^ concentration and operated at a hydraulic retention time of 16–50 d, but they also added an equivalent amount of carbon as methanol to stimulate methanogenic activity. These rates on an hourly basis (0.007 and 0.060 g L^−1^ h^−1^, respectively) are 170 and 20 times slower than the ruminal value calculated above.

The above calculations of cellulose degradation rates for the mixed ruminal community raise some interesting questions. For example, how do these rates compare to those of the pure cultures of *F. succinogenes* or the ruminococci, which we regard as the predominant ruminal cellulolytic bacteria? According to Russell et al. [94], dividing the maximum specific growth rate on cellulose of 0.1 h^−1^ by the observed growth yield of 0.2 g cells (g cellulose)^−1^ translates to a specific rate of cellulose degradation of 0.5 g cellulose (g cells)^−1^ h^−1^. If the cellulolytic bacteria in vivo display the same fundamental growth properties as in pure culture, ruminal cellulolysis could be accounted for by the bacterial fraction if 23.8% of the community (=0.119/0.5 × 100%) were cellulolytic. This percentage is several-fold that of the abundance of known cellulolytic species from 16S-based sequencing or qPCR-based abundance assessments (Table 1). This suggests some combination of at least two possibilities: (i) some cellulolytic members of the bacterial community have remained hidden within the culture-independent assessments (i.e., are cellulolytic but have not been identified as such); and, (ii) a substantial fraction of ruminal cellulose degradation is due to the eucaryotic community (protists and fungi); the latter possibility is in accord with some of the studies described in Section 3.2 and Section 3.3 above. 

What accounts for the more rapid rate of cellulose hydrolysis by ruminal communities, compared to other anaerobic habitats? As noted in Section 2 above, rumination is likely is a major contributor to the high rate and extent of cellulose degradation. And, as noted in Section 3.4, in vitro rates of cellulose degradation by both pure and mixed cultures of ruminal bacteria are proportional to available substrate surface area, and the measured rates of cellulose degradation in these in vitro cultures (in which rumination is absent) are due in part to the use of finely divided pure MCCs such as Avicel or Sigmacell. Unfortunately, direct comparisons to cellulose degradation rates by non-ruminal microbes is complicated by the fact that these latter studies rarely describe the particle sizes, properties or sources of the cellulose used. Regardless, it is clear that rumination is a unique co-treatment process that has no parallel in anaerobic digesters, aquatic sediments or other anaerobic habitats.

A second factor that likely contributes to the superior cellulolytic properties of ruminal communities is the more complete adherence of ruminal microbes to the cellulose. As noted from studies of pure cultures (see Section 3.1 above), adherence to cellulose is required for rapid cellulose hydrolysis by ruminal bacteria and fungi. The requirement for adherence is also indicated from in vitro studies with mixed ruminal bacteria, in which (i) cell adherence to cellulose particles and rate constant of cellulose degradation both decreased in parallel with decreasing pH [77], (ii) inhibition of adherence by methylcellulose inhibits cellulose degradation [39] and (iii) removal of adherent cells immediately arrests further cellulose degradation [40]. By contrast, adherence to cellulose is rarely described in anaerobic digesters. An exception is the study of Vavilin et al. [87], who noted that cellulose was degraded by a cellulose-adherent bacterial population in an anaerobic digester operated at short (2.97 d) solids retention time (SRT); however, another anaerobic digester operated at 13.7 d SRT displayed both slower cellulose hydrolysis, and lack of adherence of bacteria to the cellulose. Interestingly, this study also reported that at longer SRTs, first-order kinetics did not adequately describe cellulose degradation, suggesting that the digester may have been at least partially limited by the low abundance of adherent cellulolytic microbes. O’Sullivan et al. [95] examined AD of MCC using an enriched culture from a landfill leachate, and noted that first-order kinetics was not followed when the cells had not fully colonized the cellulose particles during a very long (4 d) initial lag period, or late in the fermentation when the biofilm on the cellulose particles had become multilayered and complex, and cellulose degradation had slowed considerably. 

An additional reason for the more rapid rate of cellulose degradation lies within the unique nature of the ruminal process and the rapid turnover of rumen contents. Because the ruminal fermentation evolved to nourish the host animal, fermentation is incomplete and results in the production primarily of VFA, compounds of high potential energy that are absorbed through the rumen wall or passed down the GI tract for absorption and further host metabolism. By contrast, other anaerobic habitats achieve complete mineralization of cellulose to methane and CO_2_, and this requires the participation of proton-reducing acetogens (which convert C_3_-C_6_ VFA to acetate, CO_2_ and H_2_) and aceticlatic methanogens (which convert acetate to methane and CO_2_). Members of these two groups are absent from the rumen (they grow too slowly to prevent their washout), but they are present in digesters, aquatic sediments, and other low-turnover habitats, where they complete the mineralization of organic matter. Inhibiting these groups results in VFA accumulation and acidification of the habitat, which restricts cellulose degradation (a well-known example being the “sour digester”). Under normal conditions, without this inhibition, VFA do not accumulate, but only because the cellulose degradation rate is rather slow, and does not outpace VFA catabolism and methanogenesis. This interpretation is consistent with the thermophilic AD work of Jeithanipour et al. [96], who showed that increasing the rate of cellulose degradation by using more rapidly degradable amorphous or pretreated cellulose as substrate decreased the organic loading rate that permitted digester operation without initiating product inhibition. In effect, the slow rate of downstream mineralization pathways under normal AD seems to have selected for a slower upstream degradation of cellulose to keep the overall digestion process in balance.

### 3.6. Hemicellulose Degradation

Hemicelluloses are a large class of non-cellulose polysaccharides typically found either in association with plant cell walls, or as storage carbohydrates in seeds or tubers. They vary widely in monosaccharide composition and linkage patterns, and as a result vary in biodegradability [97]. They are typically named according to the composition of their main backbone chains and the pendant groups attached to the main chain (e.g., arabinoxylans are chains of β-1,4-linked xylose that have arabinose as pendant sugars). Recently there has been a trend to narrow the definition of hemicelluloses to β-(1→4)-linked polysaccharides having an equatorial configuration at C1 and C4, thus excluding polysaccharides like β-(1→4)-galactans, in which the configuration is axial [98].

Generally, hemicelluloses have received less attention than cellulose as substrates for ruminal biodegradation, owing to their more complex structure, the inconsistency of their commercial availability, and their chemical behavior. For example, many hemicelluloses in purified form exhibit considerable but incomplete solubility in water, which complicates quantification (e.g., in measuring their disappearance gravimetrically). Most hemicelluloses are relatively soluble in strongly alkaline aqueous solution, and labile in strong acid; this latter property has been exploited in detergent extraction protocols used to separately quantify NDF and ADF [99]. These two fractions are used as proxies for fiber when formulating ruminant dietary rations.

Early studies of ruminal hemicellulose degradation utilized mixtures of hemicelluloses extracted from specific forages or crop residues, typically by solubilization in aqueous alkali followed by precipitation in ethanol. More recent studies have focused on individual hemicelluloses isolated from plant tissues (often seeds), in which these particular hemicelluloses are highly enriched (e.g., linear xylans from tobacco stalks [100], or xyloglucan from tamarind seeds [101]). Across this broad range of substrates, it is apparent that hemicellulose degradation by ruminal microbes is fundamentally different from cellulose degradation (Table 4).

#### 3.6.1. Bacterial Hemicellulose Degradation

Because of the great variation in composition and structure of hemicelluloses, one would expect that the hemicellulose-degrading community would be abundant and diverse. In fact, ruminal hemicellulose degradation is carried out by two distinct physiological groups: the highly specialized cellulolytic bacteria (described in Section 3.1 above) and certain nutritionally versatile non-cellulolytic bacteria.

It has long been known that the predominant ruminal cellulolytic bacteria are highly capable at hydrolyzing both pure hemicelluloses, and the hemicellulose component of intact forges. Dehority [102] isolated hemicelluloses from five different sources (flax, corn hulls, oat hulls, alfalfa and fescue), then examined their degradation (hydrolysis) and utilization (fermentation of the hydrolytic products) by eight pure strains of predominant ruminal cellulolytic bacteria. All strains were able to convert ethanol-insoluble hemicellulose to ethanol-soluble-carbohydrate (indicative of hydrolysis), but only three strains (*R. albus* 7, *R. flavefaciens* C94, *R. flavefaciens* B1a) were able to utilize the hydrolytic products for organic acid production or cell growth, and only the last strain was able to utilize monomeric xylose. Substantial differences were observed in the extent of degradation of the substrates, with flax hemicellulose being most degradable and corn hull hemicellulose being the most resistant. Coen and Dehority [103] subsequently examined degradation and utilization of hemicelluloses within intact forages (bromegrass and alfalfa) harvested at different stages of maturity, by a large number of pure ruminal strains. The same cellulolytic strains that effectively degraded and utilized the pure substrates showed generally similar activity on the hemicelluloses in the intact forage.

Hemicellulose hydrolysis by the cellulolytic specialists is consistent with analysis of their genomes. *F. succinogenes* contains 36 genes encoding hemicellulases (predominantly xylanases), although some of these annotations are weak (i.e., have low homology to the CAzyme database), and have not yet been supported by enzymological studies with authentic hemicellulose substrates. In *R. albus* 7, expression of several putative xylanases is upregulated during growth on cellulose [42]. *R. flavefaciens* produces numerous novel CBMs that recognize not only β-glucans, but also β -galactans and homogalacturonan; these CBMs likely support degradation by catalytic components of its highly complex cellulosome [104].

In characterizing the hemicellulolytic capabilities of the specialist cellulolytic bacteria, Coen and Dehority [103] also identified strains of the non-cellulolytic *Butyrivibrio fibrisolvens* as robust hemicellulose degraders, consistent with their preferential isolation from rumen fluid on xylan-amended medium [105]. This metabolic generalist is also capable of protein hydrolysis, and a few strains are also cellulolytic, though weakly so. As in the case of the cellulolytic specialists, less hemicellulose was removed from the more mature than from the less-mature forages, confirming the strong role of matrix interactions among cell wall components as an important determinant of biodegradability [83]. 

*Prevotella,* the most abundant genus of bacteria in the rumen [17], includes numerous species capable of hemicellulose degradation and utilization [105,106]. Four species of *Prevotella* (*P. bryantii, P. brevis, P. ruminantium* and *P. albensis*) were found to degrade and ferment xylan, arabinoxylan, 4-O-methylglucuronxylan, xyloglucan, β-glucan, and glucomannan, but less completely than did *B. fibrisolvens*. However, enrichment cultures from ruminal fluid obtained from cows fed hemicellulose-rich diets were dominated by *Prevotella* (and secondarily by *Succiniclasticum*) strains when the same group of xylans were substrates, and by *Streptococcus* when glucomannan was the substrate [106], suggesting that these latter taxa may be more important in rumine. More recently, in vitro incubations of ruminal fluid with corncob xylan have demonstrated the enrichment of the genera *Prevotella, Selenomonas, Lactobacillus, Bifidobacterium* and *Atopobium* [107], although parallel metagenomic analysis did not identify these genera as major contributors to the xylanase activity of the enrichments. 

The ruminal degradation of xylans (particularly arabinoxylans and 4-O-methylglucuronosylans) has received special attention, owing to their great abundance in forages. Because the degradation of the substituent groups from forages occurs more rapidly than does the degradation of the xylosyl component, it appears that prior removal of these side chains is essential for more complete degradation of the main chain [108]. Once the side chains are removed, hydrolysis of the main chain is accomplished by a mixture of exo- and endo-xylanases, most of which are extracellular and not cell-bound [109]. Hydrolysis primarily generates xylooligosaccharides (xylodextrins), analogous to the hydrolysis of cellulose to cellodextrins, and as is the case for cellulose degradation, xylodextrins are often utilized more readily than is the xylose monomer [110].

#### 3.6.2. Hemicellulose Degradation by Eucaryotes

Unlike the fairly broad distribution of cellulolytic capability among some (mostly ciliated) protists, xylan-degrading capability seem much more narrowly distributed, mostly within the genera *Eudiplodinium* and *Polyplastron* [59]. Another ciliate protist, *Epidinium ecaudatum,* has been reported to engulf particles of xylan, though much less frequently than for MCC, and cell extracts exhibited xylanolytic activity [65]. Genomic analysis indicates that the protists have a relatively small number of GH genes for hemicellulose degradation [14]. 

Ruminal fungi, particularly *Neocallimastix* and *Orpinomyces,* are thought to be highly hemicellulolytic, based on the abundance of particular classes of GH enzymes in their genomes, although pure culture studies using defined hemicelluloses are lacking. Recent metagenomic studies have shown correlations between a number of protist (*Entodinium, Diploplastron*, and *Eudiplodinium*) and fungal (*Neocallimastix, Orpinomyces*, and *Olpidium*) genera and the xylanases produced in enrichment cultures of ruminal fluid with corn cob xylan [107].

### 3.7. Kinetics of Hemicellulose Degradation

From a quantitative standpoint, hemicellulose degradation has not received the same attention as has cellulose degradation. Nevertheless, a few generalities have emerged. Pure hemicelluloses are usually degraded relatively rapidly, and the degradation appears to follow first-order kinetics. Complications arise when hemicelluloses are present in matrices with other plant cell wall components. In lignin-free synthetic composites of cellulose and linear xylans, produced by growing the cellulose-synthesizing bacterium *Acetobacter xylinum* in the presence of linear xylan from tobacco stalks, the two components were degraded at similar rates, k = 0.12 h^−1^ [111]. Because the pure xylan was degraded much more rapidly (k = 0.52 h^−1^) by the same inocula, it appears that the rate of xylan degradation in the composite was due to its limited accessibility when surrounded by cellulose. Despite this, both components were ultimately degraded completely, in the absence of lignin. 

Most studies of hemicellulose degradation by ruminal microbes have used intact forages as substrate, usually dried and ground to a moderate particle size to facilitate work at small (laboratory) scale. In these studies, degradation of different hemicelluloses has been estimated from rates of degradation of specific component monosaccharides (e.g., xylan removal is estimated from xylose disappearance, even though smaller amounts of xylose are also present in other hemicelluloses such as xyloglucan). A universal observation from such studies is that, in contrast to cellulose, hemicellulose removal from forages is much slower than is the degradation of pure hemicelluloses. The residual feedstock is enriched in xylosyl units (xylan) relative to glucosyl units (cellulose) due to the former’s more intimate association (not only physically, but also via covalent bonds), with lignin, the indigestible phenylpropanoid polymer in the plant cell wall. 

### 3.8. Microbial Interactions

#### 3.8.1. Interactions among Cellulolytic Bacteria

The fact that cellulose degradation is limited by available surface area of the cellulose suggests that cellulolytic microbes are continuously engaged in competition for this limiting substrate. It is of some interest, then, to examine how the different cellulolytic species interact to deal with this limitation.

The similarities in strategies for cellulose degradation among the predominant ruminal cellulolytic bacteria, outlined in Section 3.5, raise two questions: How abundant are the individual cellulolytic species in the rumen, and what determines this abundance? The first question has largely been examined by culture-independent quantification, particularly using species-specific qPCR or whole-community 16S rRNA gene sequencing of bulk community DNA. The former method has been more widely used due to its species specificity, while the latter method is used more generally for broad assessments of community composition across higher taxonomic levels (usually phylum, family and genus). The qPCR studies agree that, at least in dairy cows, *F. succinogenes* and *R. flavefaciens* have roughly similar abundance, while *R. albus* is much less abundant (Table 1).

The second question has been approached in defined mixed culture studies, in which the relative abundance of the different species is interpreted within the context of known physiological features of the individual species. Early studies by Odenyo et al. [112,113] revealed that in binary (two-strain) batch cultures using either cellobiose or acid-swollen cellulose as substrate, *R. albus* 8 and *F. succinogenes* S85 co-existed at similar abundance; *R. flavefaciens* FD-1 displayed a slight advantage over *F. succinogenes*; and *R. albus* suppressed *R. flavefaciens* (Table 5). Ternary culture of all three strains resulted in co-existence of *R. albus* and *F. succinogenes*, but again a rapid disappearance of *R. flavefaciens*. The authors proposed that *R. albus* produced an agent that inhibited growth of *R. flavefaciens*. Chan and Dehority [114] also observed inhibition of *R. flavefaciens* by *R. albus* culture supernatants, and determined that the active agent was a protein. Chen et al. [115] later purified one such compound, albusin B, which displayed the properties of a class III bacteriocin, and whose inhibition appeared specific only to strains of *R. flavefaciens*.

To examine interactions of the cellulolytic bacteria under more stringent conditions, Shi et al. [116,117] compared binary culture of different strains in grown on either MCC or cellobiose, under either substrate-excess batch cultures or substrate-limited continuous culture (but substituting *R. albus* strain 7 for strain 8). Batch culture results were generally similar to those of Odenyo et al., although the *R. albus/R. flavefaciens* co-cultures displayed variations in which the latter sometimes outnumbered the former, and in other cases was eclipsed. Under substrate limitation (continuous culture) on cellobiose, either of the ruminococci excluded *F. succinogenes*, and *R. albus* again excluded *R. flavefaciens*. Exclusion of *F. succinogenes* by the ruminococci was likely due to the latter’s known poorer affinity for products of cellulose hydrolysis determined separately in pure culture [53]. In continuous culture on cellulose, *F. succinogenes* was excluded by *R. flavefaciens*, but retained a numerical abundance over *R. albus*. Surprisingly, a coculture of the two ruminococci species displayed a greater abundance of *R. flavefaciens*, similar to results occasionally observed in batch culture. These results suggest that a threshold abundance of *R. albus* was required to allow production of a critical concentration of the inhibitor, beyond which *R. flavefaciens* was excluded. Bacteriocin production by *R. albus* apparently serves to prevent exclusion of this species by *R. flavefaciens*, which displays otherwise superior growth properties, including a more rapid rate of cellulose hydrolysis, higher growth yield, lower maintenance coefficient (Table 2), as well as higher affinity toward cellodextrins [53]. *R, flavefaciens*, meanwhile, has been shown to produce a soluble inhibitor of growth that allows it to dominate *F. succinogenes* in coculture [118], but the inhibitory agent has not been identified.

Production of inhibitory agents has also been reported for hemicellulose utilizers. In particular, many strains of *Butyrivibrio fibrisolvens* produce bacteriocins that inhibit the growth of other strains of this species, with varying levels of specificity [119].

#### 3.8.2. Interactions between Cellulolytic and Non-Cellulolytic Bacteria

Owing to the central role of cellulose in ruminant nutrition and great diversity of sugar utilizing microbes in the rumen, the number of potential interactions between these two groups is likely to be large. One of the best studied examples involves *F. succinogenes* and the sugar-fermenting bacterium *Selenomonas ruminantium* [120]. This latter species can convert succinate to propionate, but cannot obtain energy from this conversion. Coculture of the two species resulted in crossfeeding of cellodextrins (released by *F. succinogenes*) to *S. ruminantium*, thus permitting growth and allowing ancillary conversion of succinate to propionate–the major precursor for gluconeogenesis in the ruminant host. In fact, cellodextrin utilization may be widely distributed among non-cellulolytic bacteria [121], and some of these species may be capable of cross-feeding other nutrients to the cellulolytics, such as branched chain VFA produced from amino acid fermentation.

Non-cellulolytic bacteria can also alter the complex interactions among the predominant ruminal cellulolytic bacteria described above (Section 3.8.1). In cellobiose-limited chemostats, tricultures of *F. succinogenes, R. albus* and *R. flavefaciens* are normally dominated by *R. albus*, but inclusion of the non-cellulolytic *Selenomonas ruminantium* or *Streptococcus bovis* results in dominance of the culture by *R. flavefaciens*, apparently due to *R. albus* being unable to sustain a sufficient abundance to benefit from its production of its *R. flavefaciens*-specific inhibitor [118]. In tricultures of the cellulolytic bacteria on cellulose at different dilution rates, *F. succinogenes* represented about two-thirds of the cellulolytic population, and *R. albus* almost all of the rest, with *R. flavefaciens* accounting for only a few percent. This proportion was not changed by inclusion of *Streptococccus bovis*, but inclusion of *Selenomonas ruminatium* shifted the cellulolytic population to about two-thirds *R. albus*, at the expense of *F. succinogenes*. While the mechanisms underlying such shifts are unclear, it is apparent that non-cellulolytic bacteria can alter population dynamics among the cellulolytic bacteria, even on pure cellulose and cellobiose. Interactions among these bacteria on actual plant cell walls (multicomponent and architecturally complex) are certain to be even more complex.

Sugar fermenting bacteria can also influence the course of hemicellulose degradation. For example, coculture of cellulolytic bacteria that hydrolyzed (but did not utilize) hemicelluloses with cellulolytic or non-cellulolytic bacteria that did utilize the hydrolytic products increased total hemicellulose utilization [104]. As has been demonstrated in ruminal cellulose degradation, xylan degradation involves extensive cross-feeding of oligomers produced by xylanolytic bacteria such as *B. fibrisolvens*, to xylodextrin-utilizing but non-xylanolytic bacteria such as *Selenomonas ruminantium* [122]. 

#### 3.8.3. Interactions among Cellulolytic Eucaryotes

Morgavi et al. [64] have reported that cellulose degradation and VFA production by the ruminal fungus *Piromyces* sp. strain OTS1 was decreased during coculture with ruminal ciliate protists in vitro, likely due to protist predation of fungal zoospores; however, the negative effects on cellulolysis may have been partially counteracted by the fibrolytic activity of the protists. 

#### 3.8.4. Inter-Kingdom Interactions

Production of antifungal agents by ruminal bacteria was first observed from pure culture experiments. Stewart et al. [123] discovered that culture supernates of both *R. albus* and *R. flavefaciens* contained a heat-labile factor that inhibited cellulose fermentation by *Neocallimastix frontalis*, but did not affect growth of the fungi on glucose; the authors suggested the agent or agents acted by preventing adherence of the fungi to cellulose. Bernalier et al. [124] reported that culture supernates of *R. flavefaciens* contained two proteins that inhibited *N. frontalis* cellulase activity. Taking a somewhat different approach, Dehority and Tirabasso [125] demonstrated that fungi disappeared rapidly from in vitro incubations of a ruminal inoculum on cellulose or ground alfalfa, unless antibiotics were added, in which case the fungi gradually proliferated and were able to degrade the substrates to about two-thirds the extent of the cultures without antibiotics (whose cellulolytic activity was solely attributed to the bacteria). The authors then examined how the fungi were suppressed in the presence of bacteria, and observed that inhibition—like that in the above pure culture studies—was not observed during growth on glucose. Further characterization revealed that the inhibitory agent was heat-stable and protease-resistant, suggesting that it is different from those produced by the ruminococci. Taken as a whole, it appears that, although these various agents await purification and characterization, the cellulolytic bacterial community may be capable of suppressing cellulolytic fungi, thus restricting the latter’s contribution to cellulose degradation.

One of the most well-established interactions in ruminal (and, more broadly, anaearobic) microbial ecology is interspecies hydrogen transfer between methanogenic archaea and H_2_-producing microbes; the latter include bacteria, fungi and protists. In all cases, methanogens act to maintain a low partial pressure of H_2_, whose accumulation would thermodynamically constrain further H_2_ production. The net result is a mutualistic relationship in which both partners benefit: the methanogen receives its primary energy source, while the H_2_ producer channels reducing equivalents to proton reduction, leaving organic fermentation intermediates available for production of acetate, thus generating additional ATP. Bacteria that participate in interspecies H_2_ transfer reactions are not in direct contact with the methanogens, though they may have close proximity within biofilms. By contrast, both protists and fungi produce a discrete organelle, the hydrogenosome [126,127,128], in which the fermentative intermediate pyruvate is oxidized to acetate, CO_2_ and H_2_. The H_2_ is released from the organelle, where it is utilized by methanogens adherent on the hydrogenosome surface. Interspecies H_2_ transfer and its mechanistic underpinnings have been clearly demonstrated in defined mixed culture of methanogens with bacteria, fungi and protists, but the relative importance of this mutualism for the different groups in the rumen remains unclear, and is likely to be influenced by other factors and other interactions. For example, based on the poor correlation between ciliate protist and methanogen abundances, it has been proposed that ciliate protists decrease methane production by predation of methanogens [129].

#### 3.8.5. Ecological Implications

Because microorganisms, including those of the rumen, follow basic ecological principles, deciphering the complex interactions during fiber degradation in the rumen can likely be informed by drawing parallels to other microbial habitats.

As noted above, the rates of both cellulose and hemicellulose degradation appear to be substrate-limited, and one would then expect that negative interactions (such as competition and amensalism) would dominate over positive interactions (such as mutualism). Indeed, both competition for substrate and production of antagonistic agents appear to dominate interactions among ruminal fibrolytic microbes (Section 3.8.1 above). This pattern is in accord with accumulating evidence across the microbial world that—at least when considering organisms at the same trophic level, on which almost all these studies have been based—(i) microbial interactions are primarily negative, and cooperative interactions are rare; (ii) cooperative interactions become even less common in habitats of high species diversity; and (iii) negative interactions have a net effect of stabilizing the community [130,131]. Among ruminal microbes, negative interactions even extend below the species level, as shown by the production of bacterioicins by *Butyrivibrio fibrisolvens* that inhibit other strains of the same species [119]. It is only when one considers the contributions of organisms at different trophic levels does one see the emergence of important positive interactions that impact ruminal fiber degradation, such as interspecies H_2_ transfer (Section 3.8.4 above). Another example involves cross-feeding, by cellulolytic bacteria, of cellulose and hemicellulose hydrolysis products to metabolic generalists such as *Prevotella*, in exchange for branched-chain VFA produced by the latter’s peptide and amino acid fermentation [2].

The rumen harbors at least three bacterial species highly capable of cellulose degradation (Section 3.1 above), and many more capable of hemicellulose degradation. Additionally, most bacterial species in the rumen have until now resisted isolation and cultivation, but metagenomic studies have identified many more species whose genomes encode fibrolytic enzymes, and the same can be said for the ruminal eucaryotes (fungi and protists). Overall, this number of species vastly exceeds the number of “degradation points” (soluble substrates plus hydrolysable substrate linkages in biopolymers) subject to enzymatic attack. Simple arithmetic thus dictates that, on average, multiple species should be able to exploit any given degradation point [132]. By this standard, the rumen fibrolytic microbiota display a robust functional redundancy that combines a diversity of enzymatic capabilities distributed among many species with a strong convergence of primary metabolism, as indicated by relatively similar proportions of VFA end products across a wide range of fermentation inocula and environmental conditions [133]. These subtle differences in niche overlap allow the many species to eke out a living from the array of available substrates, without having to engage in the pure and simple competition that restricts community diversity. Ultimately, this niche overlap is traceable to unique collections of fibrolytic enzymes encoded in the genomes of individual fibrolytic strains.

This functional redundancy, however, may not be absolute, at least for fiber degradation. As noted in Section 3.2 above, the contribution of protists in ruminal fiber degradation appears to be quantitatively significant, but its disruption is followed by incomplete niche replacement, (i.e., other microbial groups—bacteria and fungi—do not appear to completely fill the void of decreased fiber degradation in defaunated animals). The same applies to fiber degradation by bacteria and by fungi, suggesting that each of these major microbial groups has mechanisms of cellulose and/or hemicellulose degradation that are unique to their specific group. 

### 3.9. Potential for Industrial Exploitation of Ruminal Fiber Fermentation

Bioconversion of abundant and renewable cellulosic biomass has long been touted as one of the cornerstone technologies for transitioning to a sustainable bioeconomy. For economic viability, cellulosics conversion require high loading rates, rapid substrate utilization, high product yields and high product tolerance, along with effective downstream processing of product mixtures.

Robert Hungate, the father of anaerobic microbiology, was the first to recognize the potential of the rumen microbial community for industrial bioconversion of cellulosic feedstocks:

“In summary, an industrial cellulose fermentation might be profitable if the cost of collection of raw materials could be minimized through the use of numerous small plants, if the small plants could be cheaply constructed, if the operation could be made automatic to decrease necessary personnel, and if the concentration of cellulose fermented could be increased by continuous removal of fermentation products. Although such a situation is at present quite out of the question as an industrial process, it is almost an exact specification of the ruminant animal, a small fermentation unit which gathers the raw materials, transfers it to the fermentation chamber, and regulates its further passage, continuously absorbs the fermentation products, and transforms them into a few valuable substances, meat, milk, etc. To these advantages must be added the crowning adaptation: the unit replicates itself.” [134].

These prescient observations can now, some seven decades later, be examined in a slightly different context—use of the microbial community, outside the rumen (i.e., “extraruminally”), in bioreactors whose design is informed by the unique characteristics of the ruminant host. As noted in this review, ruminal cellulose degradation is a complex, highly integrated and evolutionarily mature process. Its superior rate and extent of cellulosics degradation, relative to non-ruminal communities, degenerates substantially as the community is separated into its component parts. This argues for the use of the entire ruminal community for any industrial fermentation. 

Critical to the use of rumen microbial communities in a biorefinery context is maintenance of the mixed culture in the bioreactor for extended periods, and stable storage of mixed cultures for later use (eliminating a frequent need for fresh inocula). Both requirements appear to be easily met. Storage of ruminal inocula at −80 °C in 5% dimethylsulfoxide yields fully functional communities upon revival [135], and long-term in vitro fermentations have been maintained on highly fibrous substrates (mature switchgrass plus distillers dried grains) for 2 years [136]. In the latter study, although the community diverged dramatically in species composition over time (including rapid loss of the protist and fungal components), they nevertheless retained most of their functionality with respect to substrate utilization and total fermentation product formation, although fermentation product ratios did change with the community over time.

#### 3.9.1. Carboxylate Production 

The natural products of the ruminal fermentation are VFAs, thus positioning the community as a promising exemplar of the carboxylate platform [137]). This platform was originally proposed and developed using mixed microbial communities from other organic-rich habitats (anaerobic digesters [138] or aquatic sediments [139]); the VFA products have a variety of industrial end uses, and are also potential precursors for chemical conversion to liquid fuels [137]. Most studies of the carboxylate platform have used readily fermentable substrates or waste materials as feedstocks, owing to the poor cellulolytic capabilities of communities from these habitats. They have also required inhibition of methanogenesis (by low pH or addition of inhibitors such as iodoform) to facilitate VFA accumulation.

The ruminal community displays several advantages over inocula from other habitats that make them strong candidates for the carboxylate platform [11]. Like other degradative communities (e.g., in anaerobic digesters or aquatic sediments), it is able to degrade almost any non-lignin organic compound, but it also has a superior capability for fermentation of cellulose and hemicellulose—compounds whose abundance make them centerpieces for the large-scale bioconversions that will be required for meaningful impact in replacing hydrocarbon-based fuel and chemical production. Second, the ruminal community is well-adapted to high loading rates (solids content in the rumen is typically in the 15% range on a DM basis). Third, unlike digester or sediment-derived communities, the ruminal community is naturally adapted to high concentrations of VFA (typically 100–200 mM). Fourth, the ruminal community displays long-term in vitro stability necessary for an industrial process. A major challenge is to improve the VFA product profile of the ruminal community (in particular, enhanced chain elongation of C_2_–C_4_ VFA to the more energetic and desirable C_5_–C_8_ acids). Once properly assembled and adapted, rumen-derived communities can likely benefit from cultivation in bioreactors designed to mimic certain aspects of the host ruminant, such as biomimetic rumination [10], or inline recovery of fermentation products [140]. 

#### 3.9.2. Ethanol Production

In contrast to their superb capacity to produce VFA, ruminal microbes are unlikely to contribute to industrial conversion of cellulosic biomass to ethanol, one of the cornerstone products of the fermentation industry. Few known ruminal microbes (and only one known cellulolytic, *R. albus*) produce significant amounts of ethanol, and only do so at relatively low yield in pure culture; moreover, these microbes show poor ethanol tolerance. Ethanol production by mixed ruminal communities (e.g., upon slug dosing of starch) is transient, and the ethanol serves as a reductant for elongation of C_2_–C_4_ VFA to caproate [141]. Direct addition of ethanol to mixed ruminal bacteria in vitro stimulates methanogenesis [142]. Collectively, these observations indicate that ethanol accumulation by mixed communities is minimal, and known ruminal ethanologens appear to lack the capability or robustness necessary for ethanol production in a biorefinery.

#### 3.9.3. Microbial Fuel Cells

Simply put, microbial fuel cells (MFCs) use microbial metabolism to convert chemical energy into electrical energy. The microbes couple the oxidation of various reduced chemicals (electron donors) on an anodic electrode surface, to the electron accepting capabilities of a (nonliving) cathodic surface, to generate an electrical current through an external circuit [143]. Conventional fuel cells have almost exclusively relied on soluble organic compounds or their fermentation products as the primary energy source. Fuel cells based on cellulosic materials provide an opportunity to use abundant feedstocks that are not in competition with human foods, for renewable energy production. 

Rismani-Yazdi et al. [144] were the first to demonstrate cellulose-driven electric current generation in MFCs mediated by ruminal microbial communities as the biocatalyst. The MFC consisted of graphite electrodes in a divided cell (an anodic and a cathodic chamber separated by a proton-exchange membrane). The anaerobic anodic chamber contained the ruminal inoculum and growth medium; the cathodic chamber contained potassium ferricyanide as electron mediator, and was maintained under aerobic conditions to permit reduction of O_2_ at the cathode. Electric current was generated immediately upon the addition of ball-milled cellulose (the electron donor), and was sustained for 2 months with periodic cellulose feeding. A maximum power density of 55 mW m^−2^ of electrode and a steady state current of 0.47 mA were attained. The maximum power density was similar to that observed in other studies with soluble substrates using nonruminal inocula, suggesting that cellulose hydrolysis (which was mediated by both the anode-adherent and planktonic communities) was not the rate-limiting step in current generation.

Wang et al. [145] have reported that a MFC based on wheat straw fermentation by ruminal inocula generated high levels of VFA (>100 mM) and an electrical current, although maximum power density was very low (0.003 mW cm^−2^). The authors attributed these low values to use of a more recalcitrant form of cellulose; a shorter incubation time (7 d) that may have restricted microbial colonization of the anode; and an accumulation of VFA. Cellulose consumption was not quantified in either study, so it is not possible to calculate power yield per unit cellulose consumed. Nevertheless, these studies provide a proof of concept for MFCs based on ruminal cellulolytic ruminal communities, that can serve as a baseline for future work.

#### 3.9.4. Bioaugmentation of Anaerobic Digesters

If, in fact, rumen microbial communities have fiber-degrading properties superior to those of anaerobic digesters (Section 3.5 above), these communities should be able to improve AD of lignocellulosics through bioaugmentation if (i) the AD process is limited by the rate or extent of lignocellulosic digestion; and (ii) the ruminal microbes can persist and thrive in the AD environment. Several attempts at this have been described within the past few years, aided in part by using molecular techniques to follow AD community dynamics before and after addition of either pure or mixed cultures of ruminal microbes, including direct inoculation of ruminal contents (see the recent review by Basak et al. [146]). This concept builds on earlier studies in which ruminal inocula by themselves were shown to rapidly degrade cellulose-containing wastes in digesters run under typical AD operating conditions (permissive temperature and pH, and long retention times) [147,148].

Overall, these more recent studies indicate that augmentation with ruminal fluid into anaerobic digesters operated at rumen-like temperatures (36–41 °C) dramatically improves methane yields—often by 2-fold or more. The effects are greatest late in the time course of digestion, suggesting that improvements are in the extent, rather than the initial rate, of digestion, consistent with reports that cellulose digestion during AD follows first-order kinetics. However, because first-order kinetics may not be universal in AD under all operating conditions (see Section 3.5), improved AD upon bioaugmentation may reflect an increased rate of fiber degradation. 

In bioaugmented AD, methane yields of 300 L per kg of volatile solids are not unusual, well above those in the rumen itself (where methane is a junior product to VFA). However, direct comparisons of fiber degradation in augmented AD reactors to that of the rumen itself are difficult because most studies have focused on biomethanation (i.e., biogas yield) and have not described the type of substrate, or its amount (initially or at various stages of digestion). Additionally, these studies have used partially filtered ruminal fluid as inoculum, from which the bulk of the fibrolytic species (being fiber-adherent) would have been removed; this would unintentionally underestimate the potential improvement from bioaugmentation.

Despite the paucity of comparative data, a few generalizations can be made. Ruminal fluid does appear to outperform (often dramatically) other inocula in improving overall methane yield, and some rumen fluid-augmented reactors can transiently accumulate VFA to concentrations typically found in the rumen itself, before the desired conversion to biogas is attained [149], suggesting that the ruminal microbes can persist in the short term. Improved AD upon bioaugmentation has been ascribed to some combination of increased diversity of CAzymes in the community and increased adherence to cells to particulate substrates—two hallmarks of ruminal fiber degradation (Section 3.1 above). The better results with ruminal inocula over cattle manure appear to be due to the great differences in community composition between the two inocula, particularly the abundance of families Fibrobacteriaceae and Prevotellaceae (effective degraders of cellulose and hemicellulose, respectively; Section 3.1 and Section 3.6.1 above), as well as in the types of methanogens [150]. Interestingly, a more targeted focus on classical ruminal bacterial species using qPCR has revealed that *F. succinogenes* had a major role in enhancing AD at 36 °C while *R. flavefaciens* and *R. albus* were more important in AD at 41 °C [149], highlighting an additional fundamental difference between the two genera. 

The potential benefits of bioaugmentation may also be realizable by use of ruminal fungi, whose addition has been reported to increase biogas yield from manures and crop residues by up to one-third [151]. Prospects for routine improvement of AD by ruminal bioaugmentation rest largely on the ability of the ruminal microbes to persist in the digester, i.e., integrate themselves into the complex AD microbial community [146]. This requirement recapitulates, in many respects, the difficulties in modifying the microbial composition of the rumen by addition of pure or mixed cultures of both ruminal and nonruminal origin [132].

### 3.10. Knowledge Gaps and Research Opportunities

#### 3.10.1. Undiscovered Cellulolytic and Hemicellulolytic Species 

Most of our knowledge of cellulolytic ruminal microbes comes from study of a few widely distributed isolates, along with “minor” species that are either no longer in culture or not readily available. Based on calculations (Section 3.5 above), it is highly likely that the ruminal community (most of whose members are as yet uncultivated) contains undiscovered cellulolytic and hemicellulolytic species. Enrichment, isolation and characterization of such species should be a high-priority research goal. How might this problem be approached?

Zehavi et al. [152] have addressed the disparity between cultivated and uncultivated species of ruminal bacteria in a broader context (i.e., not focused on the fibrolytic population). They engaged in a systematic cultivation effort that employed (i) highly enriched media with multiple sugar substrates, with and without relatively large amounts of clarified ruminal fluid to provide additional unanticipated nutrient needs; (ii) variable sample dilution prior to plating; and (iii) sequencing of 16S rRNA genes of isolates for comparison to OTUs in the original inoculum. Simultaneously varying the medium type and dilutions enhanced the total number of OTUs obtained, and yielded both abundant OTUs and a surprisingly high proportion of rare OTUs. Ultimately, the authors were able to cultivate 23% of the OTUs in the samples. This strategy should provide a roadmap for isolating novel species, and thus is worth applying toward the isolation of novel fibrolytic species. 

Another promising effort using a somewhat simpler protocol was provided by Opdahl et al. [153], who established enrichment cultures on cellulose from beef cattle ruminal fluid. Sequencing of community 16S rRNA genes revealed that each of the seven enrichment communities yielded a different OTU as its prominent member; two of these were the well-known *F. succinogenes* and *R. flavefaciens*, but the others included three undescribed species from family Ruminooccaceae and one from Candidatus phylum Saccharibacter. Thus, sequencing methods can be used to identify promising enrichments from which novel isolates might be obtained. Such efforts might even enable re-isolation of “lost” species such as the highly cellulolytic *Clostridium lochheadii* (see Section 3.1.3).

Studies focused on the enrichment and isolation of novel cellulolytic and hemicellulolytic species can include culture-independent efforts to identify new species (and their fibrolytic enzymes) using widely available metagenomic techniques. A pioneering example of such an approach is the work of Hess et al. [154], in which an intensive metagenomic investigation of microbes adhering to a switchgrass substrate incubated within the rumen of a single cow yielded 268 GB of microbial DNA that encoded 27,755 putative genes for carbohydrate-active enzymes, along with 15 assembled genomes of novel uncultured microbial species. Once whole bacterial genomes are reconstructed from complex gut metagenomes, it becomes possible to devise novel enrichment strategies to isolate targeted species for further characterization, as has been shown for the use of a starch/urea/bacitracin medium for the isolation of Succinivibrionaceae strain WG-1 from the Tammar wallaby [155]. Similar approaches are likely to be successful for undiscovered cellulolytic and hemicellulolytic species.

#### 3.10.2. Quantitative Aspects of Cellulosics Degradation by Ruminal Eucaryotes

Both fungi and protists appear to have unique roles in cellulose (and possibly hemicellulose) degradation within the rumen, and much has been learned about the relevant enzymes produced by each group. However, our knowledge of the quantitative aspects of degradation remains minimal. Data are needed regarding kinetic order, rate constants, extent of digestion, as well as how these are affected by environmental conditions such as pH. In pursuit of such data, it would be particularly useful to use the same types of substrates that have been used in bacterial degradation studies, to permit direct comparison with the data from those bacterial systems. Additional efforts should also be made to develop culture methods that allow the fibrolytic ruminal eucaryotes to persist in vitro, to allow comparison with pure bacterial in vitro cultures with respect to substrate utilization and product formation.

#### 3.10.3. Overcoming the Recalcitrance of Cellulosic Substrates

Despite the relatively rapid rates of fiber degradation achieved with ruminal inocula, a certain proportion of the polysaccharides in plant cell wall material remains undegraded, not only upon passage from the rumen, but even after prolonged incubation in vitro [156]. This is almost certainly due to matrix interactions between certain polysaccharides (likely one or another xylan) and lignin, but the specific covalent linkages in these “indigestible residues” remain unidentified. In low-turnover habitats such as soils, the degradation rate of holocellulose (i.e., cellulose plus hemicellulose) in plant residues eventually declines over time to that of lignin [157], and the asymptotic values reached vary among plant species, consistent with known interspecific differences in composition and linkage patterns. Fuller characterization of the most slowly degraded linkages present just prior to reaching the point of indigestibility may reveal targets for enrichment or selection of novel microbes from these habitats, that may be useful in directed evolution or bioaugmentation studies to improve the extent of cellulosics degradation in extraruminal bioreactors, if such changes could be stably incorporated into the microbial community. 

## 4. Concluding Remarks

As major components of plant biomass, cellulose and hemicellulose are key energy sources that drive both microbial and animal heterotrophy. While the flux of cellulose and hemicellulose through degradative pathways occurs throughout nature, it is in the rumen that this degradation reaches its peak intensity in terms of specific and volumetric productivity. Rapid degradation rates are driven by microbial adherence to substrate, diverse (and sometimes unique) enzymatic strategies, and effective removal of fermentation products, assisted by the unique contribution of the host’s rumination. These fundamental features offer inspiration to improving the degradation of cellulose and hemicellulose in engineered biomass conversion venues such as waste treatment plants and biorefineries.

Apart from its importance in the global carbon cycle and animal agriculture, the rumen is also a living laboratory for microbial ecology. Fibrolytic microbes engage in fierce competition for an abundant, yet rate-limiting, substrate, and they supplement this competition with numerous examples of amensalism. These relationships are consistent with the dominance of negative interactions in nature among microbes at the same trophic level. At the same time, rumen fibrolytic microbes establish productive positive interactions with microbes at other trophic levels, especially mutualisms that enhance the growth of both the fibrolytic and non-fibrolytic partner. The status of the rumen as an ecological laboratory is enhanced by several features that make it an ideal experimental system. It is a self-contained unit whose inputs are easily controlled and whose outputs are easily measured (far superior in this regard to, for example, lakes or soils). Individual experimental units are represented by individual animals, which (i) facilitates well-replicated studies; (ii) allows generalization of observed phenomena across “habitats” (multiple rumina), and (iii) provides a platform for examining interactions (both physiological and genetic) between microbial communities and the animal host.

On the whole, then, ruminal degradation and utilization of cellulose and hemicellulose are attractive targets for continued research in both basic and applied microbiology. Such research, aided by continuing advances in genomics, ecological theory, and animal genetics, remains a vibrant and exciting field for the future.

## Figures and Tables

**Figure 1 microorganisms-10-02345-f001:**
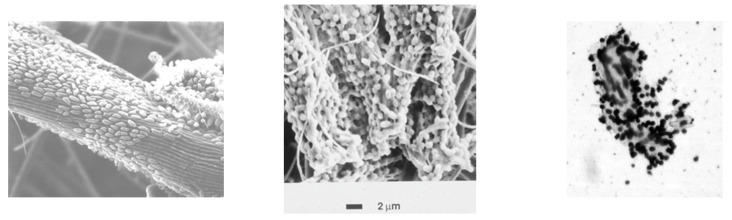
The three predominant species of ruminal cellulolytic bacteria adhering to fibers of pure cellulose. Left panel: Scanning electron micrograph of *Fibrobacter succinogenes* S85. This species adheres via pilins, fibro-slime proteins and carbohydrate-binding modules (CBMs) bound to its cell wall. Partial removal of cells during sample preparation reveals parallel grooves resulting from oriented degradation of the cellulose fiber along its crystallographic axis. Center panel: Scanning electron micrograph of *Ruminococcus flavefaciens* FD-1. Right panel: Light micrograph of crystal violet-stained *Ruminococcus albus* 7. Both species of ruminococci adhere via pilins and CBMs, and stabilize adherence by producing an extracellular glycocalyx composed of carbohydrate and protein.

**Table 1 microorganisms-10-02345-t001:** Relative abundance of predominant ruminal cellulolytic bacteria in the rumen, determined using quantitative polymerase chain reaction (qPCR).

Percentage of 16S rRNA Gene Copy Number ^a^			Notes ^b^	References
*Fibrobacter* *succinogenes*	*Ruminococcus* *albus*	*Ruminococcus flavefaciens*		
0.834	0.004	0.613	2 RC dairy cows, same diet	[17]
0.625	0.034	1.08	2 RC lactating cows; MFD induction study	[18]
0.92	0.03	1.47	2 RC lactating cows; inoculum for in vitro experiments	[19]
0.01	0.002	0.1	5 two-year old dairy cows; time-course population assessment	[13]
0.545	0.108	NT ^c^	8 RC lactating cows, MFD induction study	[20]
1.31	0.141	1.51	36 weaned dairy calves; effects of BCVFA and folic acid supplementation	[21]

^a^ Within each study, values are means across all animals and across all treatments. ^b^ BCVFA, branched-chain volatile fatty acids; MFD, milk fat depression; RC, ruminally cannulated. ^c^ NT, Not tested.

**Table 2 microorganisms-10-02345-t002:** Major differences among the three predominant ruminal cellulolytic bacteria, revealed from studies in pure culture.

Characteristic	*Fibrobacter* *succinogenes*	*Ruminococcus* *albus*	*Ruminococcus* *flavefaciens*
Phylum	Fibrobacteriota	Firmicutes	Firmicutes
Growth substrates	Cellulose, cellodextrins, glucose	Cellulose, cellodextrins, glucose, various hemicelluloses	Cellulose, cellodextrins, various hemicelluloses
Mode of adherence	Fibro-slime proteins, CBMs	CBMs, glycocalyx enriched in Glc, Xyl, Man	CBMs, glycocalyx enriched in Rha, Glc, Gal
Cellulase enzyme organization	Noncellulosomal; Surface bound via covalent linkages to cell wall	Cellusomal or non-cellulosomal depending on strain	True cellulosome, but employing single-binding dockerins
Fermentation end products	Succinate, acetate, CO_2_	Acetate, ethanol, H_2_, CO_2_	Acetate, succinate, formate, H_2_
Rate constant for cellulose degradation (h^−1^) ^a^	0.108	0.049	0.11
True growth yield (g cells [g cellulose]^−1^) ^a^	0.23–0.25	0.11	0.23–0.30
Maintenance coefficient (g cellulose [g cell^−1^ h^−1^]) ^a^	0.04–0.06	0.10	0.05–0.07

^a^ Growth data determined from cellulose-limited continuous culture studies for *F. succinogenes* S85 [27], *R. albus* 7 [28], or *R. flavefaciens* FD-1 [29].

**Table 4 microorganisms-10-02345-t004:** Differences in cellulose and hemicellulose degradation by ruminal microorganisms.

Characteristic	Cellulose	Hemicellulose
Substrate	Linear β-1→4 linked glucosyl units, no side chains, varying from crystalline to amorphous	Many combinations of sugars and linkages. Noncrystalline
Water solubility	Insoluble	Low to high
Localization	Mostly in secondary wall	Widely distributed in wall, sometimes abundant in non-herbaceous parts (seeds, tubers)
Association with lignin	Weak to none	Physical and chemical (covalent linkages)
Degrading microbes	Widely distributed across bacterial, fungal and protist lineages. Bacteria are nutritional specialists.	Primarily bacteria and fungi that are nutritional generalists
Degradation rate in pure form	~0.1 h^−1^	Up to 0.5 h^−1^
Degradation rate in plant cell walls	~0.1 h^−1^	0.02–0.04 h^−1^

**Table 5 microorganisms-10-02345-t005:** Outcome of competition experiments among the three predominant ruminal cellulolytic bacteria in batch (substrate-excess) or continuous (substrate-limited) culture.

Substrate ^a^	Cultivation Mode	Inoculation	Relative Abundance (%) ^b^			Ref.
			*R.* *albus*	*R. flavefaciens*	*F. succinogenes*	
ASC	Batch	*Fs* S85 + *Ra* 8	45		55	[113]
		*Fs* S85 + *Rf* FD-1		58	42	
		*Ra* 8 + *Rf* FD-1	100	ND		
		*Fs* S85 + *Ra* 8 + *Rf* FD-1	42	ND	58	
MCC	Batch	*Fs* S85 + *Ra*.7	44		56	[116]
		*Fs* S85 + *Rf* FD-1		42	58	
		*Ra* 7 + *Rf* FD-1	Variable ^c^	Variable ^c^		
	Continuous	*Fs* S85 + *Ra* 7	10.1–21.8		78.1–90.7	
		*Fs* S85 + *Rf* FD-1		>96.5	<3.5	
		*Ra* 7 + Rf FD-1	14.9	85.1		
Cellobiose	Batch	*Fs* S85 + *Ra* 8	50		50	[112]
		*Fs* S85 + *Rf* FD-1		81	19	
		*Ra* 8 + *Rf* FD-1	100	ND		
		*Fs* S85 + *Ra* 8 + *Rf* FD-1	50	ND	50	
	Batch	*Fs* S85 + *Ra* 7	28		72	[117]
		*Fs* S85 + *Rf* FD-1		49.9	50.1	
		*Ra* 7 + *Rf* FD-1	Variable ^c^	Variable ^c^		
	Continuous	*Fs* S85 + *Ra* 7	>99.8		<0.17	
		*Fs* S85 + *Rf* FD-1		>99.8	<0.17	
		*Ra* 7 + *Rf* FD-1	>99.8	<0.14		

^a^ ASC, acid-swollen cellulose, MCC, microcrystalline cellulose. ^b^ based on 16S rRNA probes. ^c^ highly variable results, in which one or the other species almost completely dominated the other.

## Data Availability

Not applicable.

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
