# Peer review of "Degradation of Cellulose and Hemicellulose by Ruminal Microorganisms"

_microorganisms, 2022, doi:10.3390/microorganisms10122345_

Round 1

Reviewer 1 Report

This review evaluates a very meaningful and big picture of the fiber (Cellulose and Hemicellulose) -degrading community of the rumen. This topic is useful in applied microbiology and knowledgeable for researches. 

Major comments:

1)    As the development of high-sequencing techniques such as metagenomics and metatranscriptomics, and binning analysis, these new methods with bioinformatics analysis might be helpful for the discovering new microbes and microbial fiber degraded genes. Thus, suggest to include this in 3.10.1.

Detail comments:
Line 98: suggest delete the “(Jami et al., 2013; Moraïs 98 and Mizrahi, 2019)”

Line 119: more than one point.
Line 270: should have a point after [51]

Author Response

Comment: As the development of high-sequencing techniques such as metagenomics and metatranscriptomics, and binning analysis, these new methods with bioinformatics analysis might be helpful for the discovering new microbes and microbial fiber degraded genes. Thus, suggest to include this in 3.10.1.

AU: I thank the reviewer for pointing out this omission from the manuscript. A paragraph, with two supporting references, has been added to include this approach (L1041-1053).

Minor edits:

L98: “(Jami et al., 2013; Moraïs and Mizrahi, 2019)” deleted as requested.

L119: Extra period at end of sentence deleted.

L270: Period inserted after “[51]”.

Reviewer 2 Report

Dear Author, your review is fine.  Thank you very much indeed.

Best regards.

Author Response

Comment: Dear Author, your review is fine.  Thank you very much indeed.

AU: I thank the reviewer for the favorable assessment.